# Counterfactual Occlusion-Aware Learning via Visibility Intervention for LiDAR Anomaly Detection

Longyu Yang [1]   Jun Liu [2]   Yap-Peng Tan [3 4]   Fumin Shen [1]   Heng Tao Shen [1 5]   Xiaofeng Zhu [6]   Ping Hu [1 4]

## Abstract

LiDAR point cloud anomaly detection is critical for autonomous system safety, yet most existing methods rely only on visible measurements, overlooking occlusion as a structured consequence of the LiDAR sensing process. We argue that anomalies are characterized not only by what is observed, but also by the spatial voids they create, which alter occlusion patterns and volumetric visibility. We propose Counterfactual Occlusion-Visibility Anomaly Learning (COVAL), a framework that intervenes on volumetric visibility during training. Using physics-conformed synthetic anomaly construction, COVAL generates paired factual and counterfactual observations with identical scene geometry but different occlusion patterns. Then, we introduce two complementary objectives: Visibility-Variant Counterfactual Reconstruction, which models occlusion-induced missing regions, and Visibility-Invariant Counterfactual Consistency, which enforces stable representations across visibility changes. Together, these objectives isolate anomaly-induced structural missingness and in turn refine representation of normal scenes, thus improving anomaly sensitivity at test time. Experiments on the standard LiDAR anomaly segmentation benchmark show that COVAL achieves state-of-the-art performance.

## 1. Introduction

LiDAR-based anomaly detection aims to identify points or regions in a point cloud that deviate from the normal struc-

[1]University of Electronic Science and Technology of China, Chengdu, China [2]Lancaster University, Lancaster, United Kingdom [3]Nanyang Technological University, Singapore, Singapore [4]VinUniversity, Hanoi, Vietnam [5]Tongji University, Shanghai, China [6]Hainan University, Hainan, China. Correspondence to: Ping Hu <chinahuping@gmail.com>.

*Proceedings of the 43$^{rd}$ International Conference on Machine Learning*, Seoul, South Korea. PMLR 306, 2026. Copyright 2026 by the author(s).

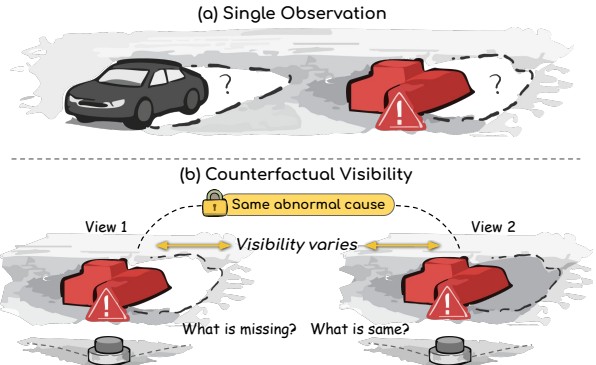

*Figure 1.* Occlusion contrast and anomaly identifiability in LiDAR sensing. (a) With a single observation, occlusion provides no contrast: missing regions caused by normal and abnormal objects are indistinguishable, limiting what can be learned about scene structure. (b) By introducing controlled synthetic abnormal perturbations while keeping normal scene structure unchanged, visibility variations reveal how abnormal occlusion disrupts otherwise consistent missing regions, in turn sharpening the model's understanding of normal objects and their occlusion patterns.

ture of outdoor environments (Cen et al., 2022; Nekrasov et al., 2025). In scenarios like autonomous driving, this capability is critical for safety, as vehicles must reliably detect unknown obstacles in complex and dynamic scenes. Unlike closed-set perception tasks such as semantic segmentation or object detection (Guo et al., 2020; Choy et al., 2019; Kong et al., 2023), anomalies in real-world LiDAR data rarely correspond to well-defined semantic categories. Instead, they arise from latent abnormal causes, such as unexpected intrusions, unusual object configurations, or structural irregularities, that perturb the scene formation and sensing process and manifest indirectly in the observed point cloud. Consequently, robust anomaly detection must transcend simple recognition of unfamiliar surface patterns; it requires reasoning about the generative process of observations. This process is fundamentally governed by ***Occlusion*** (Shan & Toth, 2009; Li & Ibanez-Guzman, 2020; Sun et al., 2020), the primary physical constraint of LiDAR sensing, which dictates that what is not seen is also informative.

Despite the critical role of occlusion in shaping LiDAR observations, most existing LiDAR anomaly detection methods learn from naturally captured point clouds and focus

exclusively on visible measurements. Learning-based approaches rely on uncertainty or reconstruction errors of observed points (Hendrycks & Gimpel, 2016; Lakshminarayanan et al., 2017), while synthesis-based methods emphasize increasing the geometric diversity of injected anomalies (Xu et al., 2025; Li et al., 2025). In both cases, the visibility structure of the scene is treated as fixed, and occlusion is largely overlooked, even though it provides object-level structural cues that can help distinguish normal scene geometry from anomaly-induced perturbations.

This limitation stems from a fundamental identifiability issue imposed by the physical constraints of LiDAR sensing (Xu et al., 2022; Balado et al., 2021). Because LiDAR operates under line-of-sight, normal scenes exhibit structured and stable missing regions determined by consistent object layouts and geometry. However, when models are trained passively on naturally captured data, each scene is observed under only a single, fixed visibility realization. Under this setting, occlusion patterns are inseparably entangled with the visible object geometry that causes them. As illustrated in Fig. 1 (a), this lack of contrast makes missing regions caused by normal and abnormal objects indistinguishable, creating an identifiability problem (Locatello et al., 2019; Creager et al., 2021). Consequently, the training data provides no opportunity to disentangle whether unobserved regions arise from normal occlusion or from abnormal structural perturbations. Even when synthetic anomalies are injected, their effects are learned only through their visible surfaces, as in existing methods, while their induced changes to surrounding occlusion patterns remain unobserved. Without access to such contrast, anomaly-induced structural missingness is absorbed into the model's representation of normal variability, leading to overly permissive notions of normality. This entanglement weakens the learning of normal scene structure and its characteristic occlusion behavior, ultimately limiting sensitivity to subtle but structurally abnormal patterns at test time.

Crucially, resolving this ambiguity requires supervision that cannot be obtained from passive single-view observations alone. Exposing the model to abnormal perturbations while controlling visibility provides a form of contrastive signal: it reveals which aspects of the scene representation should remain invariant to changes in visibility, and which missing regions cannot be explained by normal occlusion geometry. Learning from such abnormal cases therefore does not merely improve recognition of anomalies, but also sharpens the representation of normal scenes by tightening the boundary between explainable occlusion-induced missingness and structurally inconsistent deviations.

To address this issue, we propose Counterfactual Occlusion-Visibility Anomaly Learning (COVAL). Our key idea is to introduce controlled synthetic abnormal perturbations while keeping the normal scene structure unchanged, and to construct counterfactual observations that differ only in the visibility of the corresponding occluded regions. This enables direct comparison of how the same abnormal cause manifests under different occlusion conditions. We design two complementary objectives: a Visibility-Variant Reconstruction objective that models occlusion-induced missing regions, and a Visibility-Invariant Consistency objective that enforces stable representations across counterfactual views. Together, these objectives isolate anomaly-induced structural missingness and refine representations of normal scenes, thereby improving anomaly sensitivity at test time. In summary, our contributions are threefold:

- We analyze a fundamental identifiability learning issue in LiDAR anomaly detection, showing that single-view observations entangle anomaly-induced structural effects with normal occlusion patterns.

- We propose Counterfactual Occlusion-Visibility Anomaly Learning (COVAL), a novel training framework that intervenes on volumetric visibility to disentangle occlusion from scene structure by constructing counterfactual observations under controlled abnormal perturbations.

- Extensive experiments demonstrate that our approach significantly improves LiDAR anomaly segmentation performance over prior methods, achieving state-of-the-art results on the standard benchmark.

## 2. Related work

Due to the importance of safety in autonomous driving, substantial prior work has studied anomaly segmentation using 2D visual data (Bogdoll et al., 2022; Shoeb et al., 2025). These methods typically aim to identify image regions that deviate from the distribution of normal driving scenes based on visual appearance cues. Many methods formulate anomaly detection as uncertainty estimation. Some approaches rely on thresholding prediction confidence, such as maximum softmax or maximum logit scores (Hendrycks & Gimpel, 2016). Bayesian approximations, including Monte Carlo Dropout (Srivastava et al., 2014) and Deep Ensembles (Lakshminarayanan et al., 2017), further estimate uncertainty by aggregating predictions from multiple stochastic forward passes or independently trained models. Outlier exposure further improves detection by introducing proxy anomalies cropped from external datasets during training (Lin et al., 2014; Zhou et al., 2017). More recent work adopts mask-based segmentation architectures to identify unknown regions at the object or mask level. For example, RbA (Nayal et al., 2023) detects regions rejected by all known categories, while Mask2Anomaly (Rai et al., 2023)

separates known and anomalous masks using global masked attention and mask-level contrastive learning.

While anomaly detection has been extensively studied in 2D images, its extension to LiDAR-based 3D point clouds remains limited. Conventional LiDAR segmentation models (Choy et al., 2019; Graham et al., 2018; Zhu et al., 2021; Lai et al., 2023; Wu et al., 2024) are not designed to handle objects unseen during training, which poses a serious challenge for autonomous driving. Several works have explored LiDAR anomaly detection by extending open-set or out-of-distribution learning paradigms. REAL (Cen et al., 2022) introduces an open-world semantic segmentation framework using redundant classifiers and scaled proxy objects. APF (Li & Dong, 2023) introduces an adversarial prototype framework that couples learnable class prototypes with GAN-synthesized unseen features. LiON (Xu et al., 2025) formulates outlier detection as selective classification with a point-wise abstaining penalty and introduces a ShapeNet-driven synthesis pipeline. REL (Li et al., 2025) lifts road patches into object-like anomalies and models object-level anomalies via relative energy learning. While effective in practice, these methods are trained on a single LiDAR observation and primarily model visible measurements, which can limit their ability to account for visibility-related structural effects that are informative for distinguishing anomalies from normal scenes.

# 3. Method

We present a counterfactual learning framework for LiDAR anomaly segmentation that explicitly accounts for occlusion and partial observability, as shown in Fig. 2. First, we introduce the problem of LiDAR anomaly detection in 3.1. Then, we introduce physics-conformed anomaly construction to insert abnormal causes within normal scenes in 3.2. Finally, we propose counterfactual occlusion-visibility anomaly learning by constructing paired observations under different visibility realizations of the same scene in 3.3.

## 3.1. Problem Definition

We study the problem of anomaly detection in LiDAR point clouds for autonomous driving. Real-world LiDAR scans may contain abnormal causes, such as unknown objects or unusual structures, that are not represented in the training data and do not correspond to predefined semantic categories. As a result, anomaly detection is formulated as an open-set problem, aiming to identify points or regions that deviate from the distribution of normal training data. Formally, a LiDAR scan is represented as a point cloud $X = \{x_i \in \mathbb{R}^4 | i = 1, ..., N\}$, where each point $x_i$ denotes a 3D location and intensity measured by the LiDAR sensor. In closed-set semantic segmentation, the objective is to learn a model $f : X \to \mathcal{C}^N$ that assigns each point a semantic

label from a predefined category set $\mathcal{C}$. In LiDAR anomaly detection, the objective is to learn an anomaly scoring function $s : X \to \mathbb{R}^N$, which assigns higher anomaly scores to abnormal points than to normal ones.

## 3.2. Physics-Conformed Anomaly Construction

To enable counterfactual reasoning about the visibility effects induced by anomalies, we first introduce *Physics-Conformed Anomaly Construction*. We formalize this process as an intervention on the LiDAR data-generating process by injecting a latent abnormal cause. These causes are not intended to model specific semantic categories; instead, they serve as abstract, structured perturbations that deviate from the normal generative distribution (Von Kügelgen et al., 2021).

Formally, given a normal LiDAR scan $\mathcal{X}$, we sample object instances $O$ from an auxiliary object set and inject them into the scene. The auxiliary set is built from randomly sampled "thing" instances in the training data without external datasets or extra annotations. An abnormal cause $\mathcal{A}$ is instantiated by perturbing the generative factors of $O$, such as scale, orientation, surface geometry, or intensity response, yielding perturbed objects $\tilde{O} = \mathcal{A}(O)$. To prevent the model from exploiting trivial sensing artifacts, all constructions are *physics-conformed*. Inserted objects are placed at physically plausible locations, respect ground contact, and are resampled in accordance with LiDAR sensing sparsity and directionality. As a result, the perturbed objects remain consistent with the LiDAR sensing process while deviating from the normal generative factors in a controlled manner.

Crucially, because object insertion conforms to the LiDAR data-generation process, an abnormal cause can affect observations not only through newly introduced points, but also by inducing additional occlusion along LiDAR rays. In our construction, counterfactual visibility realizations are obtained by explicitly controlling these occlusion effects: for the same abnormal cause and fixed scene geometry, we selectively **keep** or **remove** points that lie behind occluding surfaces according to the LiDAR line-of-sight constraint. As a result, the underlying scene structure and abnormal perturbation remain unchanged, while only the visibility of occluded regions varies. This allows the same abnormal cause to produce multiple counterfactual point clouds that differ solely in their occlusion-induced missing regions.

## 3.3. Counterfactual Occlusion-Visibility Anomaly Learning

We adopt a counterfactual learning framework (Pearl, 2009; Kusner et al., 2017) to model visibility variations induced by abnormal causes. We denote $\nu$ as a visibility realization determined by LiDAR line-of-sight constraints. For a fixed

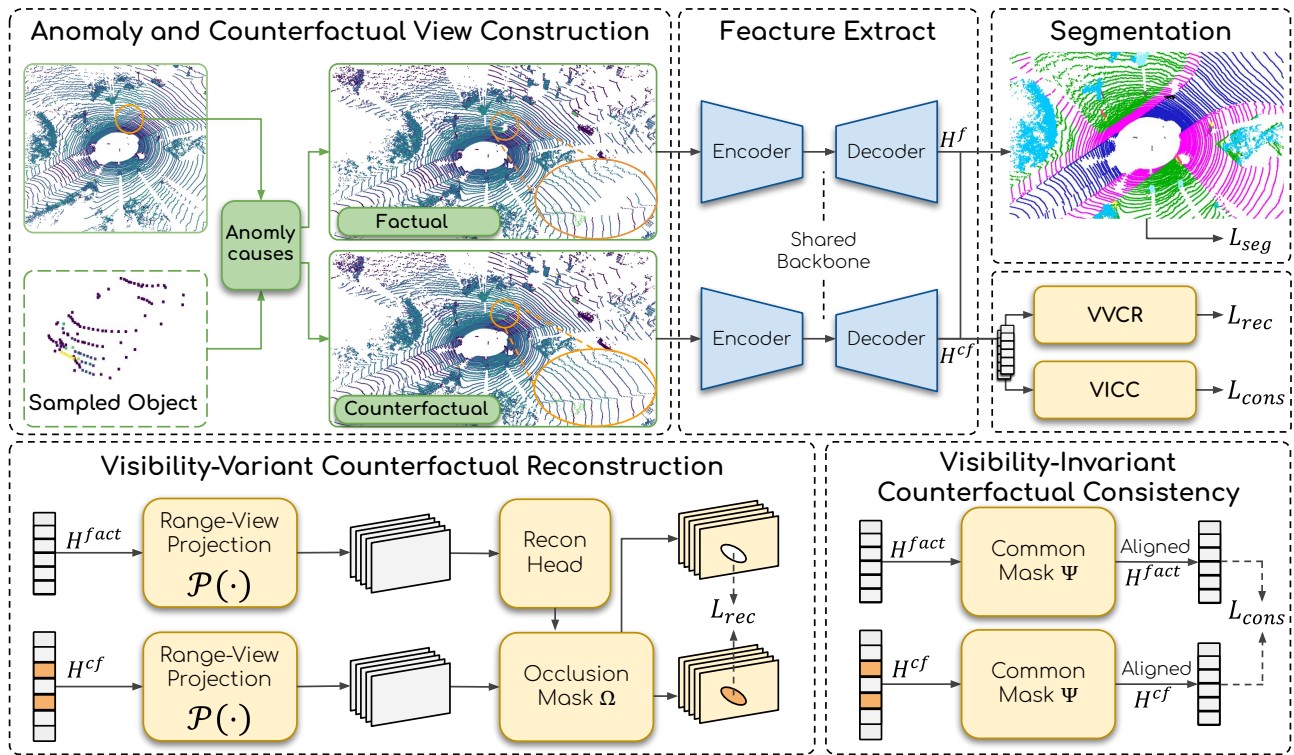

*Figure 2.* Overview of the proposed Counterfactual Occlusion-Visibility Anomaly Learning (COVAL) framework. Physics-conformed abnormal causes are inserted into normal scenes to construct paired factual and counterfactual LiDAR observations through explicit visibility intervention. Both views are processed by a shared backbone, while anomaly segmentation is performed on the factual view. During training, two counterfactual objectives are applied: *Visibility-Variant Counterfactual Reconstruction (VVCR)*, which models occlusion-induced missing regions, and *Visibility-Invariant Counterfactual Consistency (VICC)*, which enforces feature consistency across commonly visible regions. Together, these objectives enable learning anomaly-sensitive representations.

underlying scene geometry $\mathcal{S}$, the observed point cloud under visibility $\nu$ is written as

$$X^{(\nu)} := \mathcal{S} \mid do(\text{visibility} = \nu), \tag{1}$$

where the intervention operates solely on the visibility mechanism. Here, the counterfactual view is used to provide controlled visibility variation under fixed scene geometry and abnormal cause, rather than to represent an additional physically captured LiDAR scan.

For each scene augmented with an abnormal cause $\mathcal{A}$, we construct a paired set of observations that differ only in *abnormal-induced occlusion*. Specifically, we form an augmented scene $\bar{X} = X \cup \tilde{O}$, where $\tilde{O} = \mathcal{A}(O)$ denotes the perturbed objects. The **factual observation** $X^{\text{fact}}$ corresponds to the physically realizable LiDAR scan, in which occlusion is enforced according to line-of-sight constraints:

$$X^{\text{fact}} = \{\, p \in \bar{X} \mid v(p) = 1 \,\}, \tag{2}$$

where $v(p) \in \{0, 1\}$ indicates whether point $p$ is visible.

The **counterfactual observation** $X^{\text{cf}}$ is obtained by intervening on the occlusion effects introduced by the abnormal

cause, while keeping the scene geometry and abnormal perturbation fixed. Concretely, points that become occluded due to the injected abnormal object are retained:

$$X^{\text{cf}} = \bar{X}. \tag{3}$$

As a result, the paired observations $\{X^{\text{fact}}, X^{\text{cf}}\}$ share identical scene structure and abnormal geometry, and differ only in occlusion-induced missing regions attributable to the abnormal cause. This construction enables direct supervision to disentangle intrinsic scene geometry from incidental visibility effects, addressing the identifiability issue discussed in Sec. 1.

### 3.3.1. VISIBILITY-VARIANT COUNTERFACTUAL RECONSTRUCTION

The factual and counterfactual views constructed above share the same scene geometry and abnormal cause, and differ only in visibility due to occlusion. This controlled difference provides counterfactual supervision: by comparing a factual observation with its counterfactual counterpart, the model is encouraged to reason explicitly about occlusion-induced missing regions of anomaly. We leverage

this property by introducing a *Visibility-Variant Counterfactual Reconstruction (VVCR)* objective.

Let $X^{\text{fact}}$ and $X^{\text{cf}}$ denote the factual and counterfactual views of the same augmented scene. Both views are processed by a shared backbone network $F(\cdot)$, producing multi-scale feature representations

$$\mathbf{H}^{\text{fact}} = \{\mathbf{H}_l^{\text{fact}}\}_{l=1}^L, \qquad \mathbf{H}^{\text{cf}} = \{\mathbf{H}_l^{\text{cf}}\}_{l=1}^L. \qquad (4)$$

Due to abnormal-induced occlusion, some background points that are present in the counterfactual view are missing in the factual view. We denote the set of such occluded points as

$$\Omega = \{\, x \in \bar{X} \mid x \in X^{\text{cf}},\ x \notin X^{\text{fact}} \,\}. \qquad (5)$$

For points in $\Omega$, no direct measurements are available in the factual observation.

For each feature scale $l$, we introduce a reconstruction head $R_l(\cdot)$ that predicts counterfactual feature representations at occluded regions using only the factual view. Since points in $\Omega$ are absent from the sparse point cloud, we impose the reconstruction constraint in the range-view domain, which provides a dense spatial grid aligned with the LiDAR scanning geometry. Specifically, sparse features are projected to range-view feature maps using a projection operator $\mathcal{P}(\cdot)$:

$$\mathbf{Z}_l^{\text{fact}} = \mathcal{P}(\mathbf{H}_l^{\text{fact}}), \qquad \mathbf{Z}_l^{\text{cf}} = \mathcal{P}(\mathbf{H}_l^{\text{cf}}). \qquad (6)$$

The reconstruction head is applied to the factual features to predict counterfactual features:

$$\hat{\mathbf{Z}}_l^{\text{cf}} = R_l(\mathbf{Z}_l^{\text{fact}}). \qquad (7)$$

The reconstruction loss is computed only at occluded locations:

$$\mathcal{L}_{\text{rec}} = \sum_{l=1}^L \mathbb{E}_{x \sim \Omega} \left[ \left\| \hat{\mathbf{Z}}_l^{\text{cf}}(x) - \mathbf{Z}_l^{\text{cf}}(x) \right\|_2^2 \right]. \qquad (8)$$

By enforcing reconstruction across multiple feature scales, the model learns to infer occlusion-induced missing regions from surrounding visible context, integrating both local geometric cues and higher-level structural information. Importantly, this objective is not intended to recover precise geometry, but to explicitly model *what is missing* due to abnormal-induced occlusion. This encourages the network to capture structured patterns of visibility disruption, rather than absorbing them into normal observation variability.

### 3.3.2. VISIBILITY-INVARIANT COUNTERFACTUAL CONSISTENCY

While visibility-variant counterfactual reconstruction models *what becomes missing* due to visibility changes, it does

not constrain how observations of the same scene should be represented across different visibility realizations. To complement this objective, we introduce a *Visibility-Invariant Counterfactual Consistency (VICC)* loss that enforces stable representations between factual and counterfactual views generated from the same scene and abnormal cause.

The factual and counterfactual views share identical scene geometry and abnormal perturbation, and differ only in visibility. Let $\mathbf{H}_l^{\text{fact}}$ and $\mathbf{H}_l^{\text{cf}}$ denote the sparse feature representations extracted at scale $l$ from the two views. Since the factual view is obtained by removing occlusion-induced points, its spatial support forms a subset of the counterfactual view. Accordingly, feature consistency is enforced only at points that are visible in both views.

We define the set of commonly visible points as

$$\Psi = \{\, x \mid x \in X^{\text{fact}} \cap X^{\text{cf}} \,\}. \qquad (9)$$

For each $x \in \Psi$, we align the counterfactual features to the factual view using coordinate-based hashing, yielding aligned features $\tilde{\mathbf{H}}_l^{\text{cf}}$ that correspond spatially to $\mathbf{H}_l^{\text{fact}}$ at scale $l$.

To enforce invariance with respect to visibility intervention, we minimize the discrepancy between aligned features across views:

$$\mathcal{L}_{\text{cons}} = \sum_{l=1}^L \mathbb{E}_{x \sim \Psi_l} \left[ \left\| \mathbf{H}_l^{\text{fact}}(x) - \tilde{\mathbf{H}}_l^{\text{cf}}(x) \right\|_2^2 \right]. \qquad (10)$$

This objective focuses on *what remains unchanged* under visibility variation. By enforcing feature consistency at commonly visible points, the model is encouraged to encode properties intrinsic to the underlying scene and abnormal cause, rather than incidental effects introduced by occlusion or partial observability.

From a causal perspective, this consistency loss can be viewed as a practical surrogate for environment-invariant representation learning (Arjovsky et al., 2019; Krueger et al., 2021), where different visibility realizations act as distinct environments. Enforcing agreement across these environments encourages representations that are invariant to visibility while remaining sensitive to structural deviations.

Together with visibility-variant counterfactual reconstruction, this objective provides complementary supervision: reconstruction models *what is missing* due to occlusion, while consistency enforces *what stays the same* despite visibility changes. This separation enables effective disentanglement of anomaly-induced structural effects from observation-specific variability under partial observability.

*Table 1.* Point-level and object-level anomaly segmentation performance with MinkNet on the validation set of STU benchmark. "OoD Data" indicates whether auxiliary out-of-distribution data are used during training. † Results obtained by our own implementation following the methodology described in the paper. $^{ML}$ and $^{AL}$ indicates anomaly score estimated via the maximum logit and learned anomaly logit, respectively.

| Method | OoD Data | ID mIoU↑ | Point-Level OOD | | | Object-Level OOD | | | | |
|---|---|---|---|---|---|---|---|---|---|---|
| | | | AUROC↑ | FPR@95↓ | AP↑ | Recall↑ | SQ↑ | RQ↑ | UQ↑ | PQ↑ |
| Deep Ensemble (Lakshminarayanan et al., 2017) | ✗ | – | 92.24 | 23.35 | 1.28 | 14.82 | 76.37 | 3.19 | 11.32 | 2.43 |
| Max Logits (Hendrycks & Gimpel, 2016) | ✗ | 63.71 | 92.43 | 22.70 | 0.91 | 7.75 | 72.75 | 0.82 | 5.64 | 0.60 |
| Void Classifier (Blum et al., 2021) | ✓ | 63.59 | 93.55 | 22.61 | 1.04 | 28.44 | 75.21 | 1.83 | 21.39 | 1.38 |
| RbA (Nayal et al., 2023) | ✗ | 63.71 | 7.49 | 99.96 | 0.02 | 0.00 | 0.00 | 0.00 | 0.00 | 0.00 |
| REAL (Cen et al., 2022) | ✓ | 58.45 | 94.02 | 23.65 | 4.18 | 2.01 | 80.49 | 2.74 | 1.62 | 2.20 |
| REL† (Li et al., 2025) | ✓ | 62.60 | 87.82 | 40.96 | 7.26 | 28.83 | 80.46 | 2.03 | 23.20 | 1.64 |
| **Ours**$^{ML}$ | ✓ | **63.75** | 98.47 | 10.90 | 82.10 | **70.63** | 80.66 | 7.57 | **56.97** | 6.11 |
| **Ours**$^{AL}$ | ✓ | **63.75** | **99.92** | **0.13** | **89.00** | 67.92 | **81.69** | **70.94** | 55.48 | **57.95** |

## 3.4. Training Objective

Our training objective combines the complementary supervision signals introduced above. Let $\mathcal{L}_{seg}$ denote the base anomaly segmentation loss, defined on the factual view using available supervision. The overall objective is

$$\mathcal{L} = \mathcal{L}_{seg} + \lambda_{rec}\,\mathcal{L}_{rec} + \lambda_{cons}\,\mathcal{L}_{cons}, \qquad (11)$$

where $\lambda_{rec}$ and $\lambda_{cons}$ balance the reconstruction and consistency terms. Together, these objectives guide the model to learn anomaly-sensitive representations that explicitly capture anomaly-induced structural missingness while remaining invariant to incidental visibility variations. The counterfactual visibility intervention is used only during training, where it regularizes the backbone to learn a tighter boundary between normal scene geometry, normal occlusion patterns, and abnormal structural missingness. This enables robust anomaly segmentation under partial observability. At inference time, the model operates on a single factual LiDAR scan without requiring counterfactual construction or visibility intervention, and anomaly scores are obtained directly from prediction confidence.

## 4. Experiment

### 4.1. Experimental Details

**Datasets.** *Spotting the Unexpected (STU)* (Nekrasov et al., 2025) is the first publicly available 3D LiDAR anomaly segmentation benchmark with three labels (inlier, outlier, and unlabeled), collected using a 128-beam LiDAR. It contains 70 anomalous sequences and two inlier sequences. Following the standard protocol, we use one inlier sequence for in-distribution training and 19 anomalous sequences for validation, without using any anomalous data during training. *SemanticKITTI* (Behley et al., 2019) is a large-scale LiDAR dataset for semantic segmentation in urban driving environments, consisting of 19 semantic classes and collected using a 64-beam LiDAR. we use sequences 00–07 and 09–10 as in-distribution training data, while sequence 08 is reserved

for ID validation. *Panoptic-CUDAL* (Tseng et al., 2025) is a panoramic LiDAR segmentation dataset collected in rainy rural driving scenarios, intended to evaluate robustness under challenging real-world conditions. It includes 19 semantic classes and is captured using a 128-beam Ouster OS1-128 LiDAR. We use five out of the six available sequences as additional in-distribution training data.

**Metrics.** Following previous work (Nekrasov et al., 2025), we evaluate anomaly segmentation performance at both the point level and the object level. For point-level evaluation, we report AUROC, FPR@95, and Average Precision (AP). For object-level evaluation, we adopt panoptic-style metrics, including Recall Quality (RecallQ), Segmentation Quality (SQ), Panoptic Quality (PQ), Recognition Quality (RQ), and Unknown Quality (UQ).

**Implementation Details.** We evaluated our approach on two backbones MinkNet (Choy et al., 2019) and Mask4Former3D (Yilmaz et al., 2024). For both backbones, we first perform pre-training using in-distribution data from SemanticKITTI, Panoptic-CUDAL, and the inlier sequences of STU to learn in-distribution representations. We then fine-tune the models using the proposed counterfactual occlusion-visibility learning framework. The weights of the visibility-variant reconstruction loss and the visibility-invariant consistency loss are set to $\lambda_{rec} = 1$ and $\lambda_{con} = 1$, respectively. During the fine-tuning stage, for Mask4Former3D, we adopt training settings similar to prior work (Nekrasov et al., 2025), with a batch size of 8 and 4 epochs. For MinkNet, we use the SGD optimizer with an initial learning rate of 0.024, momentum 0.9, and a OneCycleLR learning rate scheduler. The model is trained for 5 epochs with a batch size of 8.

**Training overhead.** Compared with the baseline MinkNet training cost of about 30 hours and 20 GB GPU memory, COVAL adds about +4 hours and +15 GB memory, mainly due to the multi-scale VVCR reconstruction heads that are removed at inference.

*Table 2.* Point-level and object-level anomaly segmentation performance with Mask4Former3D on the validation set of STU benchmark. "OoD Data" indicates whether auxiliary out-of-distribution data are used during training. ‡ Results directly reported from original paper. $^{ML}$ and $^{AL}$ indicates anomaly score estimated via the maximum logit and learned anomaly logit, respectively.

| Method | OoD Data | ID | Point-Level OoD | | | Object-Level OoD | | | | |
|---|---|---|---|---|---|---|---|---|---|---|
| | | PQ↑ | AUROC↑ | FPR@95↓ | AP↑ | Recall↑ | SQ↑ | RQ↑ | UQ↑ | PQ↑ |
| Deep Ensemble (Lakshminarayanan et al., 2017) | ✗ | – | 90.93 | 37.34 | 6.94 | 17.70 | 79.96 | 9.10 | 14.15 | 7.27 |
| MC Dropout (Srivastava et al., 2014) | ✗ | 60.72 | 65.76 | 79.32 | 0.17 | 3.54 | 74.36 | 3.48 | 2.63 | 2.59 |
| Max Logits (Hendrycks & Gimpel, 2016) | ✗ | 60.72 | 87.27 | 68.76 | 2.02 | 26.64 | 79.26 | 2.06 | 21.12 | 1.63 |
| Void Classifier (Blum et al., 2021) | ✓ | 47.97 | 89.77 | 79.50 | 2.62 | 17.35 | **81.27** | 8.98 | 14.10 | 7.30 |
| RbA (Nayal et al., 2023) | ✗ | 60.72 | 73.00 | 100.00 | 1.64 | 21.84 | 78.58 | 2.75 | 17.16 | 2.16 |
| REAL (Cen et al., 2022) | ✓ | 60.96 | 95.34 | 28.25 | 28.25 | 15.48 | 75.98 | 21.27 | 11.76 | 16.16 |
| REL‡ (Li et al., 2025) | ✓ | 59.09 | **97.85** | 9.60 | 10.68 | 49.34 | 78.89 | 13.46 | 38.93 | 10.62 |
| **Ours**$^{ML}$ | ✓ | **60.98** | 97.34 | 17.97 | **64.35** | 53.25 | 77.10 | 8.28 | 41.05 | 6.39 |
| **Ours**$^{AL}$ | ✓ | **60.98** | 97.70 | **7.33** | 62.16 | **55.16** | 77.48 | **29.01** | **42.74** | **22.47** |

## 4.2. Main Results

Table 1 reports the point-level and object-level anomaly detection performance of MinkNet on the STU validation set. Our method consistently outperforms all prior approaches across both evaluation levels, while maintaining strong in-distribution segmentation performance. At the point level, Ours$^{ML}$ already improves over existing baselines, indicating that the learned representations are highly anomaly-sensitive even when using a standard MaxLogit-based scoring strategy. Building upon this, Ours$^{AL}$ with AnomLogit, which explicitly uses the anomaly-class logit for anomaly scoring, further yields substantial improvements across all metrics. While several baselines achieve relatively high AUROC, they remain limited by high false positive rates or low precision. In contrast, our method dramatically reduces FPR@95 to 0.13 while achieving a high AP of 89.00, indicating accurate and well-calibrated anomaly localization with minimal false alarms on normal regions under the proposed learning framework and direct anomaly scoring strategy. At the object level, while using MaxLogit already improves performance, our method further shows pronounced advantages. We obtain the highest Recall, RQ, UQ, and PQ scores by a large margin, demonstrating improved detection and segmentation of anomalous object instances. This suggests that our counterfactual occlusion-visibility learning framework effectively captures anomaly-induced structural patterns, which is critical for robust object-level anomaly segmentation. Notably, our method achieves these gains without sacrificing closed-set segmentation accuracy, attaining the highest original mIoU among all methods.

Table 2 reports the OOD detection performance using Mask4Former3D on the STU validation set. Our method achieves consistent improvements across both levels metrics. At the point level, while Ours$^{ML}$ already demonstrates strong detection capability, our approach substantially reduces false positives, achieving the lowest FPR@95 of 7.33 with competitive AUROC. The large improvement in AP compared with prior methods further indicates that the pre-

*Table 3.* Ablation study on AnomSeg with MinkNet backbone. PCAC denotes physics-conformed anomaly construction, AL denotes auxiliary anomaly class logit, VVCR denotes visibility-variant counterfactual reconstruction, and VICC denotes visibility-invariant counterfactual consistency.

| PCAC | AL | VVCR | VICC | AUROC↑ | FPR@95↓ | AP↑ |
|---|---|---|---|---|---|---|
| | | | | 92.43 | 22.70 | 0.91 |
| ✓ | | | | 98.20 | 12.93 | 78.49 |
| ✓ | ✓ | | | 99.64 | 0.47 | 87.09 |
| ✓ | ✓ | ✓ | | 99.88 | 0.18 | 88.52 |
| ✓ | ✓ | ✓ | ✓ | **99.92** | **0.13** | **89.00** |

dicted anomaly scores are well calibrated and sharply localized, suggesting that the proposed counterfactual learning objectives effectively suppress spurious anomaly responses in ambiguous but normal regions. At the object level, while Ours$^{ML}$ improves recall over existing baselines, Ours$^{AL}$ achieves the best overall performance, with clear gains in RecallQ, RQ, UQ, and PQ, reflecting improved anomaly detection capabilities. Although some baselines report high SQ, this is mainly due to their lower anomaly recall, as SQ is averaged over matched instances and can be inflated by detecting only a few easy objects. In contrast, our method achieves substantially higher recall while maintaining competitive SQ, leading to superior object-level performance when evaluated jointly across Recall, RQ, PQ, and UQ.

## 4.3. Method Analysis

**Ablation on COVAL Design.** Table 3 presents an ablation study on anomaly segmentation using the MinkNet backbone, systematically evaluating the contribution of each component in the proposed COVAL framework. The baseline model, trained with standard closed-set segmentation and max-logit scoring, performs the worst, exhibiting limited separability between normal and anomalous regions and a very high false positive rate. This result confirms that naïve confidence-based scoring is insufficient for reliable anomaly detection in complex 3D scenes, where uncertainty alone cannot capture structural abnormality.

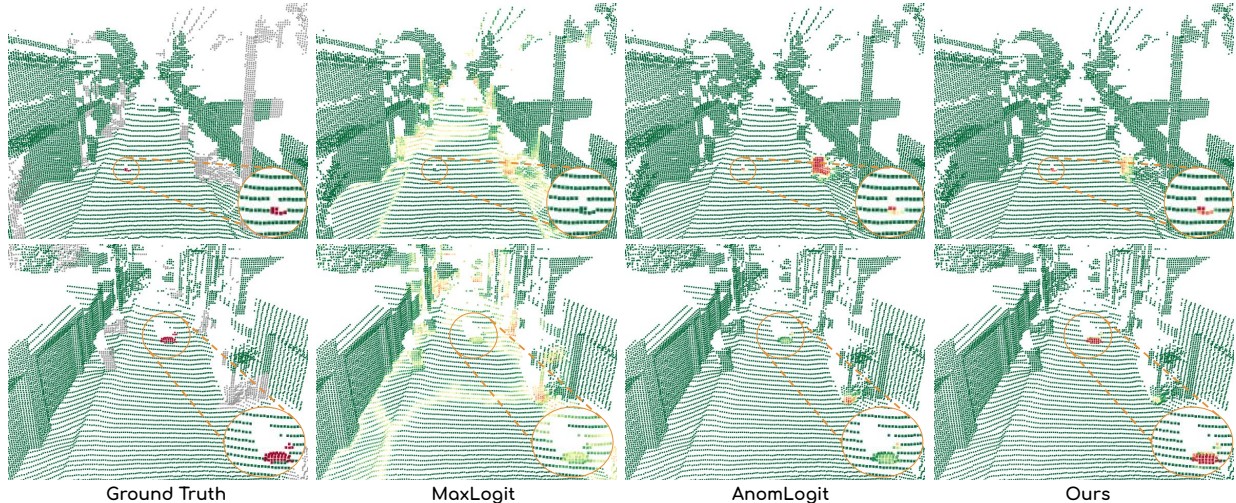

*Figure 3.* Qualitative results on the STU validation set. MaxLogit shows the weakest performance, AnomLogit (PCAC+AL) improves the results, and our full model produces the most accurate predictions.

Introducing **physics-conformed anomaly construction (PCAC)** leads to substantial performance gains across all metrics. By exposing the model to physically plausible abnormal causes during training, PCAC encourages the network to learn a more meaningful boundary between normal and abnormal structures, rather than relying on incidental appearance cues. As a result, AUROC increases significantly and false positives are markedly reduced. Further adopting the **anomaly logit (AL)** of the synthetic anomaly class as the anomaly score yields additional improvements, as it provides a more direct and calibrated anomaly signal than taking the maximum over non-anomalous logits, leading to a sharp reduction in spurious responses on normal regions.

Incorporating **visibility-variant counterfactual reconstruction (VVCR)** consistently enhances performance. Specifically, AUROC improves from 99.64 to 99.88, FPR@95 drops from 0.47 to 0.18, and AP increases from 87.09 to 88.52. These gains indicate that explicitly modeling occlusion-induced missing regions enables the network to better distinguish true anomalies from spurious responses caused by partial observations, thereby improving robustness to visibility variations and incomplete measurements.

Finally, adding **visibility-invariant counterfactual consistency (VICC)** further refines anomaly discrimination. By enforcing representation consistency across different visibility realizations of the same abnormal cause, VICC suppresses residual occlusion-induced variability that is not explained by scene structure. This leads to the lowest false positive rate (0.13) and the highest AP (89.00), yielding the best overall performance. Qualitative results in Fig. 3 further corroborate these improvements, illustrating clearer anomaly localization and reduced spurious responses.

**Control for Generic Regularization.** To further verify

*Table 4.* Ablation for generic regularization. RM denotes replacing the physics-conformed visibility intervention with random masking.

| Setting | AUROC↑ | FPR@95↓ | AP↑ |
|---|---|---|---|
| RM-Recon | 99.85 | 0.30 | 86.72 |
| RM-Recon+Cons | 99.76 | 0.30 | 87.64 |
| Ours | 99.92 | 0.13 | 89.00 |

*Table 5.* Effect of reconstruction weight $\lambda_{rec}$ without consistency loss on point-level anomaly segmentation performance.

| $\lambda_{rec}$ | AUROC↑ | FPR@95↓ | AP↑ |
|---|---|---|---|
| 0.1 | 99.85 | 0.24 | 88.77 |
| 0.5 | 99.62 | 0.50 | 86.59 |
| 1.0 | **99.88** | **0.18** | **88.52** |
| 2.0 | 99.65 | 0.49 | 87.23 |
| 5.0 | 99.82 | 0.28 | 87.21 |

that the improvement is not merely due to generic auxiliary regularization, we further replace the physics-conformed visibility intervention with random masking while keeping the same reconstruction and consistency losses. As shown in Tab. 4, random masking performs worse than our method, especially in FPR@95 and AP. This indicates that the gain comes from modeling anomaly-induced visibility structure rather than from auxiliary losses alone.

**Parameter Sensitivity.** Table 5 analyzes the sensitivity to the reconstruction weight $\lambda_{rec}$ when the consistency loss is disabled. We observe that performance follows a clear trade-off: overly small weights limit the effect of reconstruction, while excessively large weights over-regularize feature learning. A moderate value, $\lambda_{rec} = 1.0$, achieves the best balance, yielding the highest AUROC and AP while minimizing FPR@95. Table 6 further examines the consistency weight $\lambda_{con}$ with $\lambda_{rec}$ fixed. Performance remains stable

*Table 6.* Effect of consistency weight $\lambda_{cons}$ while $\lambda_{rec} = 1$ on point-level anomaly segmentation performance.

| $\lambda_{cons}$ | AUROC↑ | FPR@95↓ | AP↑ |
|---|---|---|---|
| 0.1 | 99.89 | 0.20 | 89.35 |
| 0.5 | 99.81 | 0.17 | **89.65** |
| 1.0 | **99.92** | **0.13** | 89.00 |
| 2.0 | 99.61 | 0.31 | 88.48 |
| 5.0 | 99.72 | 0.30 | 89.41 |

*Table 7.* Ablation study on multi-scale feature utilization, progressively incorporating features from high- to low-resolution scales.

| Feature Scales | AUROC↑ | FPR@95↓ | AP↑ |
|---|---|---|---|
| Highest scale | 99.85 | 0.34 | 88.22 |
| Top-2 scales | 99.75 | 0.37 | 88.30 |
| Top-3 scales | 99.71 | 0.29 | 88.35 |
| Top-4 scales | 99.87 | 0.15 | **89.45** |
| All scales | **99.92** | **0.13** | 89.00 |

over a wide range of values and peaks around $\lambda_{con} = 1.0$, indicating that the proposed consistency objective is robust and does not require careful tuning.

**Multi-scale Sensitivity.** Table 7 evaluates the impact of multi-scale feature utilization. Progressively incorporating feature scales from high to low resolution consistently improves performance, suggesting that anomaly cues arise at multiple spatial extents. High-resolution features capture local geometry, while lower-resolution features provide broader context for occlusion reasoning. The best results are achieved when all scales are used, confirming that combining fine-grained local details with coarse contextual information is critical for robust 3D anomaly segmentation.

**Sensitivity to Synthetic Anomaly Construction.** We further study whether PCAC depends on the semantic realism, category composition, or scale of injected synthetic anomalies. As shown in Table 8, even a simple cube, which does not belong to any training category, substantially outperforms REL. This indicates that PCAC does not rely on recognizing realistic semantic object shapes, but mainly benefits from structured geometric perturbations and their induced visibility changes. Using a realistic single class further improves performance, while using all "thing" classes achieves the best AP and FPR@95, suggesting that structural diversity can expose the model to a broader range of abnormal geometry and occlusion patterns.

We also vary the scale range of inserted objects in Table 9. AUROC remains stable, showing that COVAL is not highly sensitive to the exact anomaly scale. However, AP decreases for larger objects, likely because oversized insertions resemble normal structures and create more ambiguous boundaries. This suggests that moderate scales provide clearer structural deviations and more precise anomaly localization.

*Table 8.* Sensitivity to source categories of synthetic anomalies.

| Source category | AUROC↑ | FPR@95↓ | AP↑ |
|---|---|---|---|
| REL | 87.82 | 40.96 | 7.26 |
| Cube | 98.90 | 7.02 | 71.13 |
| Single class | 99.73 | 0.87 | 85.20 |
| All thing classes | 99.64 | 0.47 | 87.09 |

*Table 9.* Sensitivity to the scale range of inserted objects.

| Size range | AUROC↑ | FPR@95↓ | AP↑ |
|---|---|---|---|
| 0.1–0.3 | 99.64 | 0.47 | 87.09 |
| 0.3–0.5 | 99.83 | 0.47 | 77.93 |
| 0.5–0.7 | 99.80 | 0.70 | 75.20 |

**Failure Analysis.** Occlusion modeling is most beneficial when anomalies introduce clear visibility changes, such as distinct occlusion boundaries or coherent missing regions. Its benefit becomes weaker for very small, low-height, or far-range anomalies. For example, Seq. 142 shows a notably lower AP in Appendix Table 10 because such anomalies generate few LiDAR returns and weak occlusion changes, making both structural evidence and visibility contrast less separable from background clutter. We also observe that far-range anomalies are more challenging, as LiDAR sparsity weakens both geometric and occlusion cues.

# 5. Conclusion

In this work, we identified a fundamental identifiability challenge in LiDAR anomaly detection arising from the physical visibility constraints of LiDAR sensing: when learning from single-view observations, anomaly-induced structural effects become entangled with normal occlusion patterns. To address this issue, we proposed Counterfactual Occlusion-Visibility Anomaly Learning (COVAL), a framework that intervenes on scene visibility by introducing physics-conformed synthetic abnormal perturbations and constructing paired factual–counterfactual observations under controlled occlusion conditions. By jointly modeling occlusion-induced missingness and enforcing visibility-invariant consistency, COVAL learns representations that disentangle normal occlusion geometry from structurally abnormal deviations. Extensive experiments demonstrate that our approach significantly improves both point-level and object-level LiDAR anomaly segmentation, establishing a new state of the art on the standard benchmark.

# Acknowledgement

This work was supported by National Key Research and Development Program of China under Grant 2022YFA1004100 and National Natural Science Foundation of China under Grant 62476048,62572093.

## Impact Statement

This work aims to advance the field of machine learning by improving the robustness of anomaly detection in LiDAR-based perception systems. While the proposed method is motivated by safety-critical applications such as autonomous driving, we do not foresee any direct negative societal impacts arising from this work. As with most advances in machine learning, potential downstream applications depend on how the technology is deployed, and responsible use remains an important consideration. This paper presents work whose primary goal is to advance the field of machine learning.

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

# A. Detailed Architecture

The detailed network structure is shown in Fig. 4. MinkNet (Choy et al., 2019) is a widely used sparse convolutional backbone for 3D point cloud perception. In our implementation, we adopt the MinkNet-34C variant as the backbone network. The encoder consists of an initial stem with sparse 3×3 convolutions, followed by four stages of downsampling blocks with increasing receptive fields. Each stage contains multiple residual blocks and progressively reduces spatial resolution while increasing feature dimensionality. The decoder mirrors the encoder structure with upsampling blocks implemented using 3D sparse transposed convolutions. Features from corresponding encoder stages are connected to the decoder via skip connections, enabling multi-scale feature fusion. The final decoder output is fed into a $(K+1)$-way classifier to produce closed-set semantic predictions along with the synthetic anomaly class.

For counterfactual reconstruction, we attach a lightweight reconstruction head to each selected feature scale. As shown on the right of Fig. 4, sparse features are first projected to the range-view domain using the projection operator described in the main paper. The reconstruction head adopts an encoder–decoder architecture composed of 2D 3×3 convolutions and transposed convolutions, operating on the projected feature maps. The encoder aggregates contextual information from surrounding visible regions, while the decoder predicts feature representations at occluded locations.

The reconstruction heads are used only during training to provide auxiliary supervision for visibility-variant counterfactual reconstruction. During inference, these heads are removed and do not introduce additional computational overhead. All other components, including the backbone and segmentation head, are shared between factual and counterfactual views.

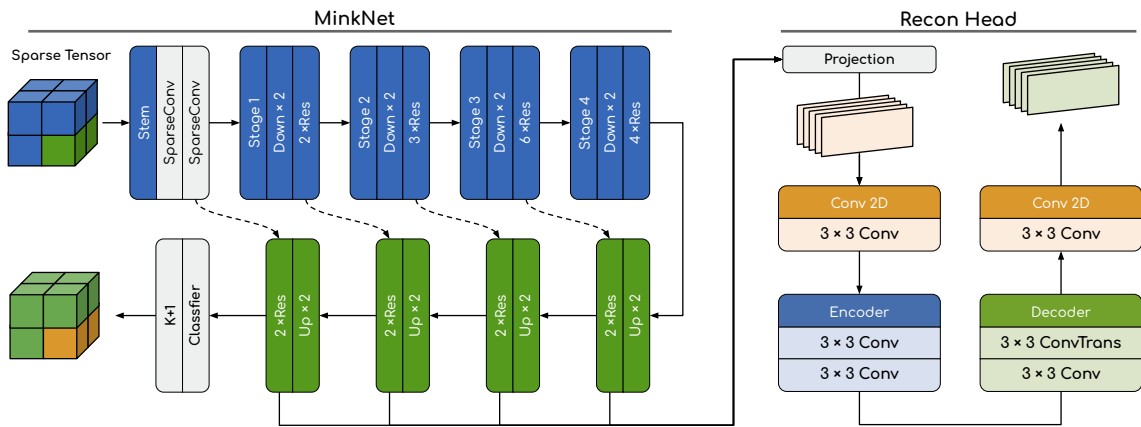

*Figure 4.* Detailed architecture of MinkNet34C and reconstruction head that we use.

# B. Detailed Performance per Sequence

We report anomaly segmentation performance on individual sequences of the STU dataset in Tab. 10 and Tab. 11. Overall, performance remains consistently high across most sequences, with only a small number of challenging cases exhibiting degraded performance, which may be due to extreme sparsity.

*Table 10.* Per-sequence point-level anomaly detection performance on the STU validation set using MinkNet (Part I).

| Seq ID | 125 | 137 | 138 | 139 | 140 | 141 | 142 | 143 | 144 | 145 |
|---|---|---|---|---|---|---|---|---|---|---|
| AUROC↑ | 99.89 | 99.99 | 99.99 | 99.98 | 100.00 | 99.91 | 98.67 | 99.97 | 100.00 | 100.00 |
| FPR@95↓ | 0.16 | 0.03 | 0.00 | 0.02 | 0.00 | 0.38 | 6.60 | 0.06 | 0.01 | 0.02 |
| AP↑ | 92.95 | 83.48 | 96.67 | 92.61 | 96.53 | 78.27 | 17.11 | 89.93 | 98.47 | 95.20 |

*Table 11.* Per-sequence point-level anomaly detection performance on the STU validation set using MinkNet (Part II).

| Seq ID | 146 | 147 | 148 | 149 | 150 | 151 | 152 | 153 | 169 |
|---|---|---|---|---|---|---|---|---|---|
| AUROC↑ | 99.99 | 99.99 | 99.99 | 99.96 | 99.99 | 100.00 | 99.99 | 99.94 | 99.68 |
| FPR@95↓ | 0.02 | 0.02 | 0.01 | 0.01 | 0.02 | 0.00 | 0.03 | 0.01 | 0.20 |
| AP↑ | 92.44 | 92.67 | 96.73 | 95.30 | 94.88 | 96.88 | 89.76 | 96.20 | 82.67 |

## C. Detailed Performance per Class

We additionally report per-class in-distribution semantic segmentation results on the SemanticKITTI validation set to verify that the proposed method does not degrade closed-set performance. As shown in Table 12, our method achieves comparable or improved performance across most semantic classes compared to prior approaches. In particular, no systematic performance drop is observed on frequent background classes or rare foreground categories, indicating that the gains in anomaly detection are not obtained at the expense of in-distribution segmentation accuracy.

*Table 12.* Per-class in-distribution semantic segmentation performance on the SemanticKITTI validation set using MinkNet backbone.

| Method | car | bi.cle | mt.cle | truck | oth-v. | pers. | bi.clst | mt.clst | road | parki. | sidew. | oth-g. | build. | fence | veget. | trunk | terra. | pole | traf. | mIoU |
|---|---|---|---|---|---|---|---|---|---|---|---|---|---|---|---|---|---|---|---|---|
| MinkNet | 96.47 | 18.05 | 61.42 | **90.70** | 61.95 | 66.55 | **85.98** | 0.00 | 93.87 | 51.78 | 81.17 | 0.00 | 91.84 | 64.63 | **89.03** | 67.52 | **77.20** | 63.04 | 49.14 | 63.71 |
| REAL | 95.29 | 8.80 | 56.62 | 53.72 | 44.06 | 62.67 | 84.19 | 0.00 | 92.66 | 41.49 | 79.17 | **0.58** | 91.07 | 61.98 | 87.50 | 66.71 | 73.11 | 62.05 | 48.87 | 58.45 |
| REL | 96.44 | 20.74 | 61.13 | 85.29 | 63.04 | 67.78 | 79.26 | 0.00 | 93.70 | 48.25 | 80.65 | 0.01 | 90.48 | 59.83 | 87.45 | 67.73 | 72.40 | **64.49** | **49.58** | 62.60 |
| Ours | **96.85** | **22.21** | **64.05** | 82.73 | **69.09** | **68.43** | 85.70 | 0.00 | **93.97** | 50.90 | **81.26** | 0.01 | **91.17** | **62.54** | 87.39 | **67.90** | 72.95 | 63.81 | 49.26 | **63.75** |

## D. Factual and Counterfactual View Visualization

As shown in Fig. 5, we visualize the factual and counterfactual LiDAR observations constructed by the proposed COVAL framework to provide an intuitive understanding of visibility intervention. Abnormal objects may occlude background structures or surrounding regions, resulting in structured and spatially coherent missing measurements in the factual view. In contrast, the counterfactual view reveals the same scene under an alternative visibility realization, exposing regions that would otherwise be unobserved.

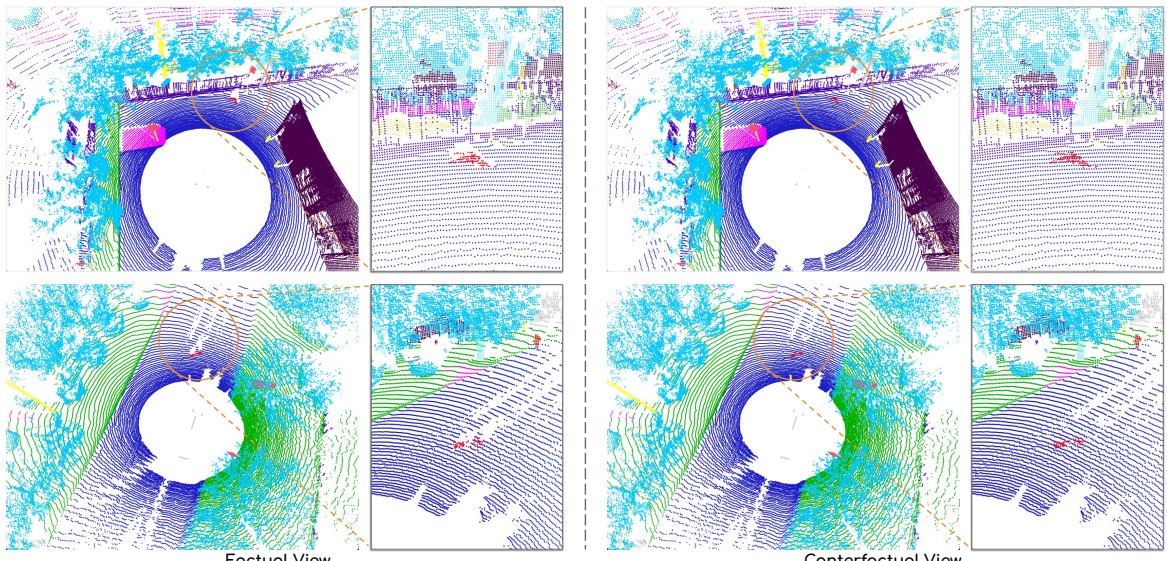

*Figure 5.* Visualization of factual and counterfactual views for same abnormal cause.

## E. More Qualitative Results

We provide additional qualitative results to further illustrate the behavior of the proposed COVAL framework under diverse scene layouts, as shown in Figures 6 to 9. We visualize anomaly segmentation outputs on a variety of STU sequences. Compared to MaxLogit-based baselines, COVAL produces anomaly predictions that are more spatially coherent and better aligned with the underlying abnormal objects, while significantly reducing spurious responses in cluttered but normal regions.

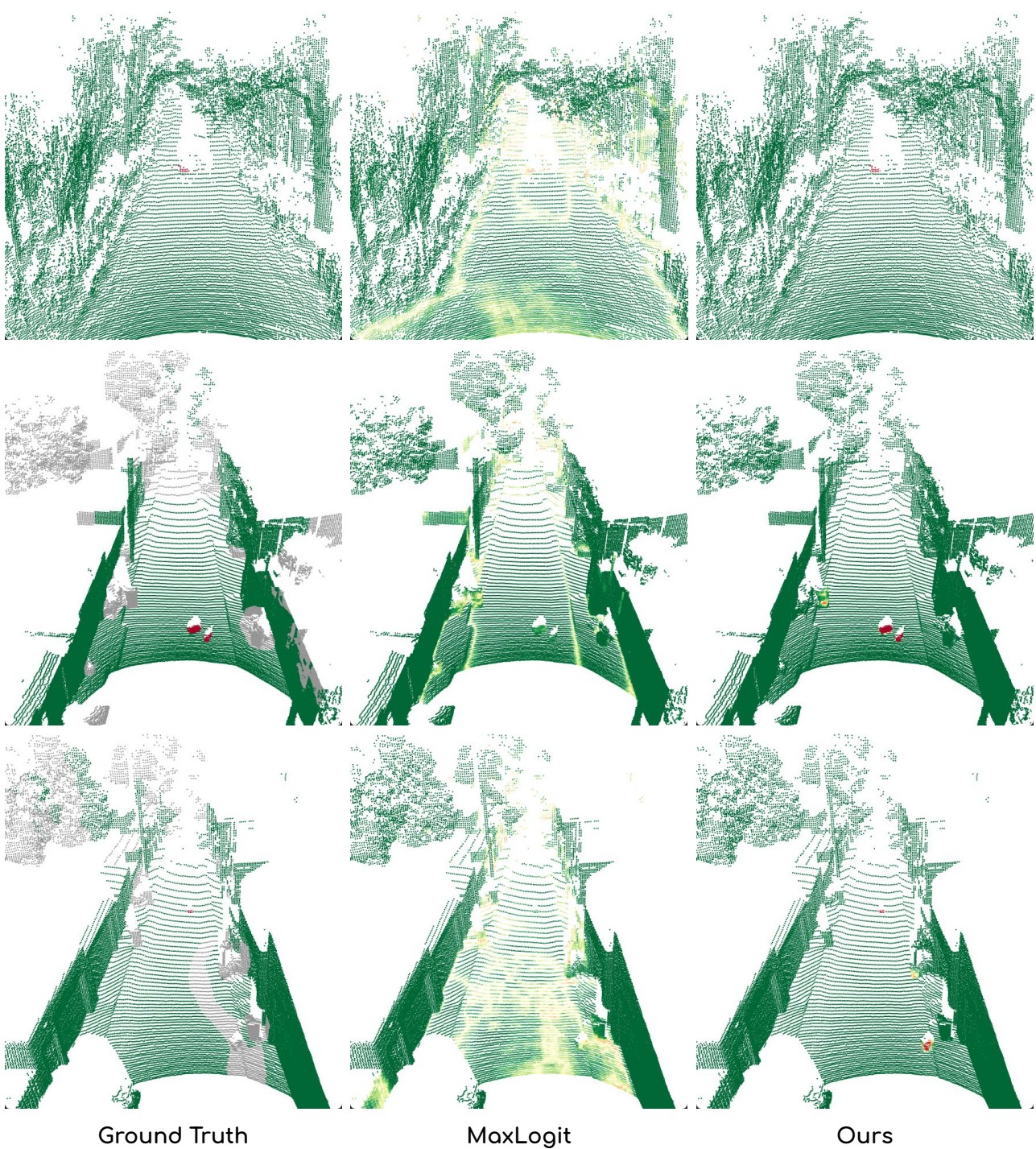

Ground Truth        MaxLogit        Ours

*Figure 6.* Qualitative results of our method on the STU validation set. Green points indicate normal points, while red points indicate abnormal points.

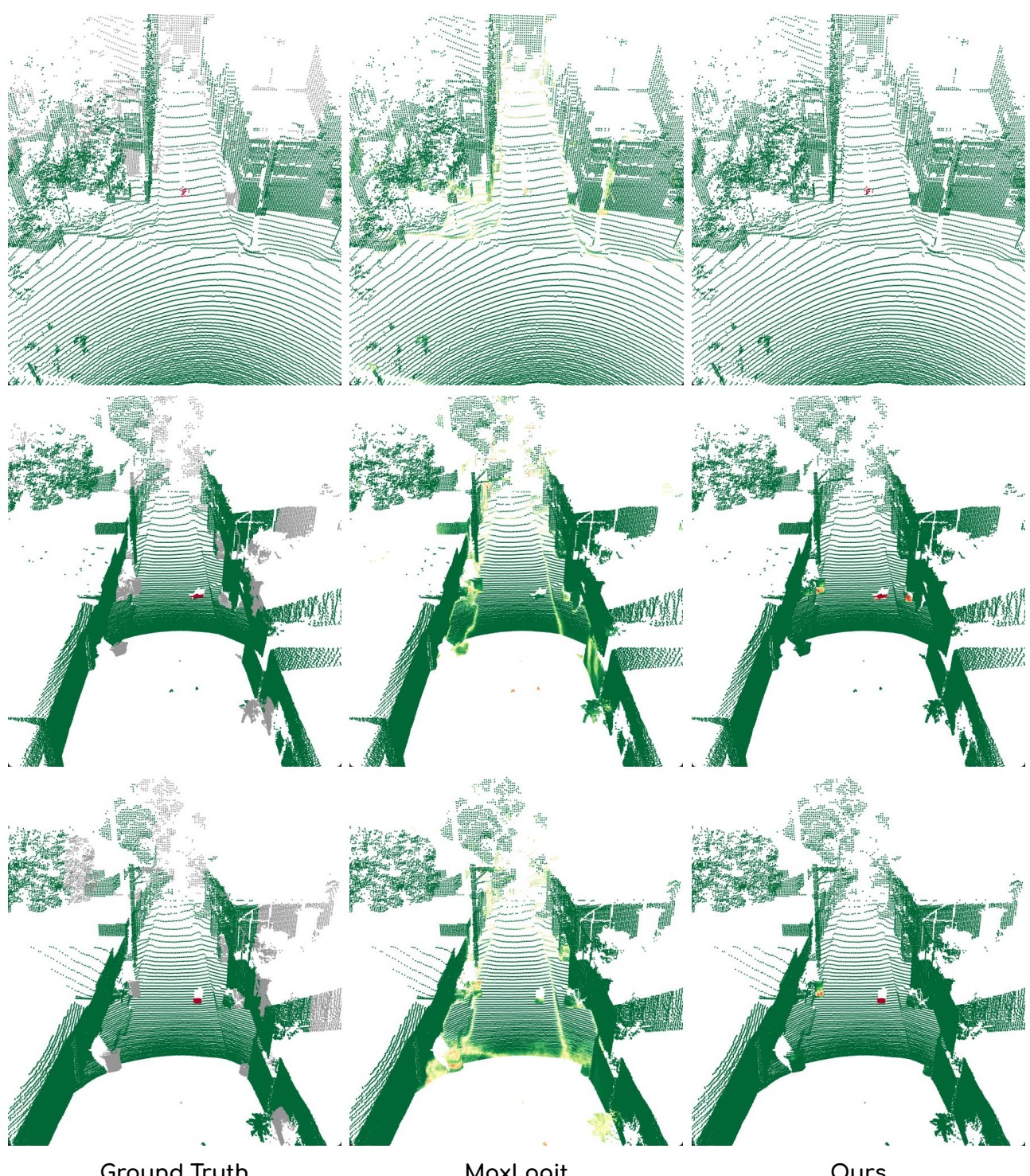

*Figure 7.* Qualitative results of our method on the STU validation set. Green points indicate normal points, while red points indicate abnormal points.

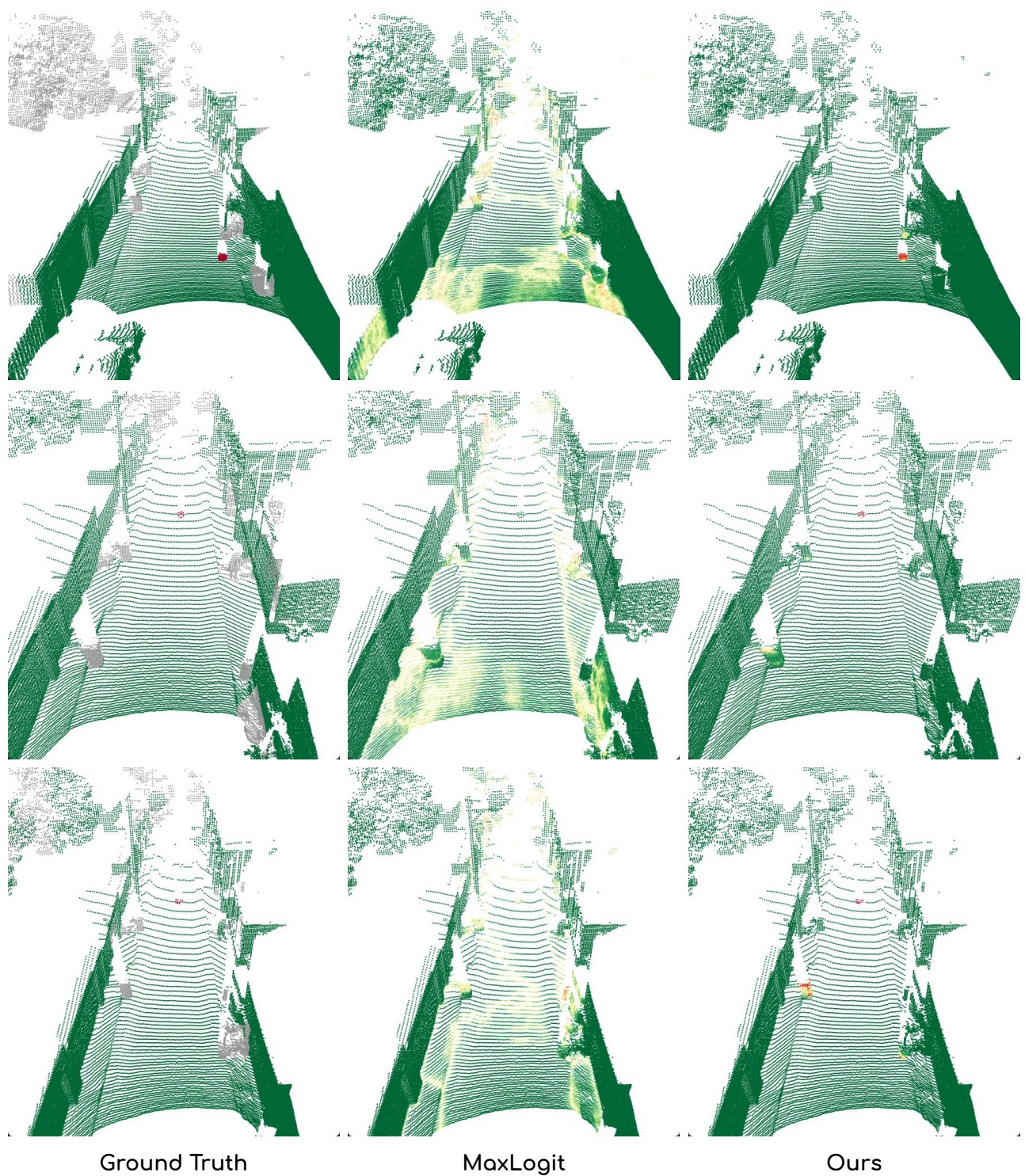

Ground Truth        MaxLogit        Ours

*Figure 8.* Qualitative results of our method on the STU validation set. Green points indicate normal points, while red points indicate abnormal points.

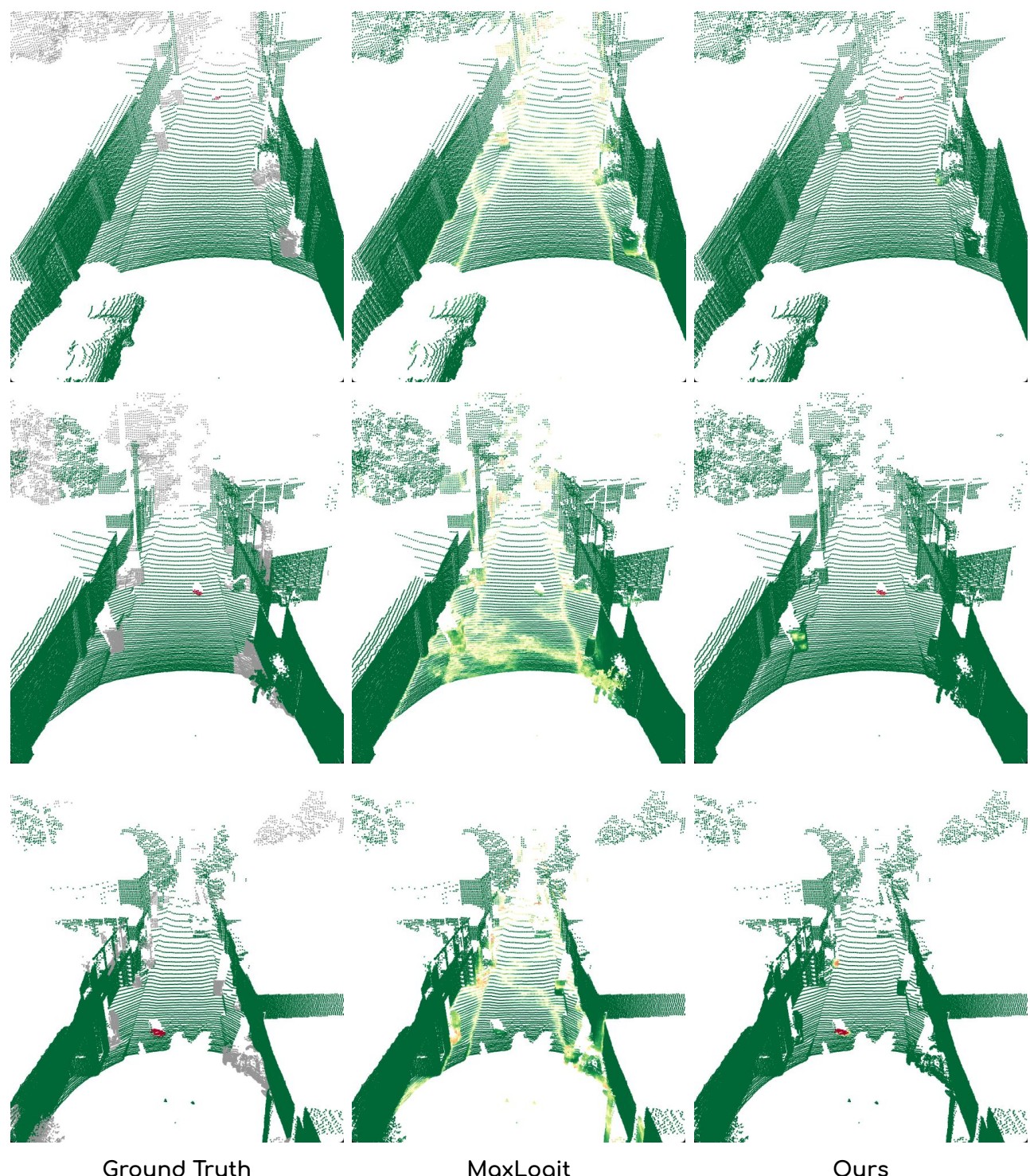

Ground Truth          MaxLogit          Ours

*Figure 9.* Qualitative results of our method on the STU validation set. Green points indicate normal points, while red points indicate abnormal points.

