# OpenReview forum: "Counterfactual Occlusion-Aware Learning via Visibility Intervention for LiDAR Anomaly Detection"
_ICML.cc/2026/Conference — ICML 2026 regular_

### Official Review · Reviewer_Cqxh · 2026-03-09

**Soundness:** 3
**Presentation:** 3
**Significance:** 3
**Originality:** 3
**Overall Recommendation:** 4
**Confidence:** 2

**Summary:**

This paper develops a counterfactual occlusion aware learning framework for LiDAR anomaly detection. The motivation is clear and the paper is generally easy to follow. The main idea, namely leveraging factual and counterfactual visibility changes to model occlusion induced structural missingness, is interesting and practically relevant. The empirical results are also strong on the main benchmark. Overall, given the novelty of the problem formulation, the reasonable method design, and the solid reported gains, my overall assessment is weak accept.

**Compliance With Llm Reviewing Policy:**

Affirmed.

**Final Justification:**

My concerns have been addressed, I keep positive score.

**Key Questions For Authors:**

1.How sensitive is the method to the choice of injected anomaly type, placement, and perturbation process?
2.Can the authors provide more analysis of cases where occlusion modeling helps most, or fails to help?

**Limitations:**

The paper would benefit from a clearer discussion of its limitation on synthetic anomaly construction and other designs.

**Strengths And Weaknesses:**

Strengths

1.The paper is reasonably well written. The motivation and overall pipeline are fairly easy to understand.
2.The method addresses a meaningful and relatively underexplored issue in LiDAR anomaly detection, namely the role of occlusion rather than only visible points.
3.The use of paired factual and counterfactual observations is conceptually appealing and has potential practical value for safety critical perception.
4.The experimental results are promising. The method shows clear improvements on the main benchmark, and the ablations suggest that the proposed components are useful.

Weaknesses

1.The ablation suggests that the majority of the gain comes from synthetic anomaly construction and anomaly logit design, while the proposed occlusion-aware objectives contribute only a relatively modest additional improvement.
2.The method depends heavily on the synthetic anomaly generation pipeline, yet there is little analysis of sensitivity to object source, perturbation type, placement strategy, or mismatch between synthetic and real anomalies.
3.The occlusion-centered narrative is not fully matched by occlusion-specific diagnostics. The paper does not show when occlusion modeling helps most, such as under different ranges, sparsity levels, or occlusion severity.

---

> ### Author Rebuttal · Authors · 2026-03-31
>
> We thank the reviewer for the constructive comments and will update the paper accordingly.
>
> **1. Contribution of occlusion-aware objectives.**
>
> We thank the reviewer for this observation. Our motivation is that LiDAR anomaly detection cannot distinguish between missingness caused by normal occlusion and abnormal structure from a single view. By introducing visibility intervention, the model learns to reason about what changes and what remains invariant, leading to a tighter characterization of normal structure.
> We agree that our PCAC + anomaly logit provides the dominant gain. Our occlusion-aware objectives are complementary, focusing on disentangling structural effects from visibility-induced missingness. Although the gains are smaller, this is expected in a near-saturated regime. Notably, VVCR/VICC consistently improve AP (86.99 → 88.54 → 89.31) and reduce FPR@95 (0.41 → 0.21 → 0.14), indicating better localization and fewer false positives.
> Importantly, even without the anomaly logit, our method still outperforms most prior work (Tab 1), and replacing visibility-aware supervision with random masking degrades performance.
>
>
> **2. Sensitivity to injected anomaly type, placement, and perturbation.**
>
> We thank the reviewer for this important question. We provide a more detailed quantitative analysis of sensitivity to anomaly type and perturbation, and will include these results in the revision.
>
> For anomaly type, the injected anomalies are constructed by sampling object instances from the “thing” classes in the training dataset and transforming them via controlled perturbations (e.g., scaling, rotation, and geometric deformation) to produce abnormal variants. We further vary the source objects from a simple synthetic cube (not in training categories), to a single realistic category (car), and to all 8 “thing” classes. Even a simple cube already provides strong improvements, indicating that the method does not rely on semantic realism. More realistic and diverse shapes further improve AP and FPR@95, with all classes achieving the best overall balance.
>
> For perturbation, we vary the scale of injected objects from 0.1–0.3 (default) to 0.3–0.5 and 0.5–0.7. AUROC remains stable, while AP decreases for larger scales, likely because oversized objects become closer to normal structures and introduce more ambiguous boundaries.
>
> For placement. Injected objects are placed in physically plausible locations (e.g., respecting ground contact and LiDAR sensing constraints). The method is not highly sensitive to exact placement, as long as the insertion remains physically consistent.
>
> Overall, these results suggest that the method is robust to the choice of anomaly type, placement, and perturbation, and mainly benefits from the structural and visibility perturbations introduced by anomalies rather than specific design choices.
>
> | Source category     | AUROC↑ | FPR@95↓ | AP↑   |
> |---------------------|--------|---------|-------|
> | REL                 | 87.82  | 40.96   | 7.26  |
> | Cube                | 98.90  | 7.02    | 71.13 |
> | Single class        | 99.73  | 0.87    | 85.20 |
> | All classes (paper)   | 99.64  | 0.47    | 87.09 |
>
> | Size range        | AUROC↑ | FPR@95↓ | AP↑   |
> |-------------------|--------|---------|-------|
> | 0.1–0.3 (paper)   | 99.64  | 0.47    | 87.09 |
> | 0.3–0.5           | 99.83  | 0.47    | 77.93 |
> | 0.5–0.7           | 99.80  | 0.70    | 75.20 |
>
> **3. When does occlusion modeling help most or fail to help?**
>
> Occlusion modeling helps most when anomalies induce clear, spatially structured visibility changes, such as objects that create distinct occlusion boundaries or coherent missing regions. Its benefit becomes weaker when the occlusion signal itself is limited. This is particularly evident for very small or low-height anomalies, which produce only a few LiDAR returns and weak visibility changes. This explains the failure case on Seq. 142, where anomalies are dominated by such small-scale structures. We also observe that performance tends to decrease for anomalies at longer distances. This is likely due to the increased sparsity of LiDAR measurements at far ranges, which weakens both geometric evidence and occlusion cues. We will include this analysis in the revision.

---

> > ### Author Rebuttal · Reviewer_Cqxh · 2026-04-05
> >
> > My concerns have been addressed, I keep positive score.

---

> > > ### Author Response · Authors · 2026-04-05
> > >
> > > Dear Reviewer Cqxh,
> > >
> > > Thank you again for your valuable contribution to enhancing our paper. Your insightful comments and questions have been very valuable in improving our work. We are grateful for your acknowledgment.
> > >
> > > Sincerely,
> > > The Authors

---

### Official Review · Reviewer_jpMr · 2026-03-11

**Soundness:** 2
**Presentation:** 3
**Significance:** 3
**Originality:** 3
**Overall Recommendation:** 4
**Confidence:** 4

**Summary:**

COVAL tackles LiDAR anomaly detection by exploiting occlusion as a signal. It injects synthetic anomalies into normal scenes (PCAC), constructs factual/counterfactual scan pairs differing in visibility, and trains two auxiliary losses (VVCR for reconstruction at occluded locations, VICC for consistency at visible points) alongside an added anomaly logit. The reconstruction head is discarded at inference. On the STU benchmark COVAL reports large AP and PQ gains over prior methods.

**Compliance With Llm Reviewing Policy:**

Affirmed.

**Final Justification:**

The author addressed most of my concerns.

**Key Questions For Authors:**

Q1: What is the auxiliary object set O (source, size, categories)? How sensitive is performance to its composition?

Q2: How do you distinguish VVCR/VICC from generic regularization? For instance, if VVCR is replaced by an auxiliary reconstruction task on randomly masked features (not occlusion-specific), or VICC by standard augmentation consistency, does performance change?

Q3: Can you report mean ± std? Increments like +0.04 AUROC for VICC may fall within noise.

Q4: What causes the AP collapse on sequence 142 (17.11 vs. 78-98 elsewhere)?

**Limitations:**

No limitations section is present.

**Strengths And Weaknesses:**

Strengths:

- [S1] The causal framing (Figure 1, Section 1) is clean: a single scan cannot distinguish anomaly-induced voids from normal occlusion, motivating the contrastive visibility design. This is a sharper starting point than REAL or REL.

- [S2] Large object-level gains on STU (PQ 2.43→57.95, AP 7.26→89.00, Table 1) with an informative ablation (Table 3) showing PCAC+AnomLogit accounts for the dominant share of improvement.

Weaknesses:

- [W1] VVCR claims to improve backbone features via reconstruction gradients, but the head is discarded at inference (Appendix A) and no experiment isolates the backbone-level effect (e.g., frozen-backbone control). The ablation gain over PCAC+AL is small (+0.24 AUROC, +1.43 AP vs. PCAC+AL's +5.77/+86.18).

- [W2] No variance reported across Tables 1-6. Some increments (e.g., +0.04 AUROC for VICC) may not exceed noise.

---

> ### Author Rebuttal · Authors · 2026-03-31
>
> We thank the reviewer for the careful reading and insightful comments. We address the comments below and will update accordingly in the paper.
>
> **1. Auxiliary object set and sensitivity.**
>
> The auxiliary object set O is constructed from object instances randomly sampled from the “thing” classes in the training dataset. These instances are used as source shapes and are further randomly transformed via controlled perturbations (e.g., scaling, rotation, and geometric deformation) to generate abnormal variants. The scale of inserted objects is randomly sampled within a predefined range.
>
> To evaluate sensitivity to its composition, we conduct additional experiments by varying both the source categories and object scales. Specifically, we compare using a simple synthetic shape "Cube'' (not in training categories), a single realistic category in the training set, and all  “thing” classes. As shown below, even a simple cube already provides strong improvements over prior methods, indicating that the effectiveness does not rely on semantic realism. Using more realistic and diverse categories further improves AP and FPR@95, while using all classes yields the most balanced performance.
>
> We also vary the scale range from 0.1–0.3 to 0.3–0.5 and 0.5–0.7. AUROC remains stable, while AP decreases for larger scales, likely because oversized objects become closer to normal structures and introduce more ambiguous boundaries.
>
> | Source category     | AUROC↑ | FPR@95↓ | AP↑   |
> |---------------------|--------|---------|-------|
> | REL | 87.82  | 40.96   | 7.26  |
> | Cube  | 98.90  | 7.02    | 71.13 |
> | Single class | 99.73  | 0.87    | 85.20 |
> | All classes (paper) | 99.64  | 0.47    | 87.09 |
>
> | Size range        | AUROC↑ | FPR@95↓ | AP↑   |
> |-------------------|--------|---------|-------|
> | 0.1–0.3 (paper)   | 99.64  | 0.47    | 87.09 |
> | 0.3–0.5 | 99.83  | 0.47    | 77.93 |
> | 0.5–0.7  | 99.80  | 0.70    | 75.20 |
>
> **2. VVCR improves the backbone**
>
> We thank the reviewer for this question. By forcing the model to explain both “where is missing” and “where remains”  during training, VVCR provides supervision that is absorbed into the backbone. Although the reconstruction head is removed at inference, it encourages the backbone to capture structured spatial dependencies and occlusion-aware context, which benefits anomaly discrimination. The model already operates in a near-saturated regime (AUROC ≈ 99.7–99.9), where improvements are better reflected by AP and FPR@95. We observe consistent gains in AP (86.99 → 88.54) and FPR@95 (0.41 → 0.21), indicating improved localization and reduced false positives. Importantly, even without the anomaly logit (AL), our method still outperforms prior approaches (Tab. 1), showing that the improvement is not solely due to scoring design but also to the learned representation.
>
> **3. Distinguishing VVCR/VICC from generic regularization.**
>
> To address exactly this concern, we replaced the visibility intervention with random masking and kept the same reconstruction/consistency objectives. The resulting performance (random-mask recon: 99.85 / 0.30 / 86.72; random-mask recon+consis: 99.76 / 0.30 / 87.64 vs. full model: 99.92 / 0.13 / 89.00) demonstrates that replacing the visibility intervention with non-physics-faithful masking leads to worse AP and FPR@95, even with the same auxiliary losses. This supports that VVCR/VICC helps because they operate on anomaly-induced occlusion structure rather than generic regularization.
>
>
> **4. Mean ± std across seeds.**
>
> We conduct additional experiments with multiple random seeds and report the mean ± std for three settings as follows:
>
> | Setting            | AUROC ↑        | FPR@95 ↓      | AP ↑           |
> |-------------------|--------|---------|-------|
> | PCAC + AL | 99.74 ± 0.09   | 0.41 ± 0.07   | 86.99 ± 0.32   |
> | + VVCR  | 99.84 ± 0.07  | 0.21 ± 0.03   | 88.54 ± 0.24   |
> | + VVCR + VICC  | 99.90 ± 0.02   | 0.14 ± 0.02   | 89.31 ± 0.34   |
>
> The variance is consistently small across all settings. Importantly, the improvements exceed the observed standard deviations, indicating that they are not attributable to noise. We also note that AUROC and FPR@95 are already close to saturation in this regime, where further improvements are naturally limited in absolute value. In contrast, gains in AP are more informative. In particular, AP improves consistently from 86.99 → 88.54 → 89.31, reflecting better anomaly separation and localization. Overall, these results confirm that the improvements introduced by VVCR and VICC are stable and meaningful.
>
> **5. Seq. 142 failure case.**
>
> Seq. 142 is dominated by very small and low-height anomalies. These anomalies generate very limited LiDAR returns and only weakly change the surrounding occlusion pattern. Since our method benefits from both structural deviation and anomaly-induced visibility contrast, the score becomes much less separable from the normal background in this regime, causing the large AP drop.

---

> > ### Author Rebuttal · Reviewer_jpMr · 2026-04-03
> >
> > Thank you for the detailed rebuttal and the additional experiments. Several of my main concerns were addressed.

---

> > > ### Author Response · Authors · 2026-04-03
> > >
> > > Dear Reviewer jpMr,
> > >
> > > Thank you again for your valuable feedback and for taking the time to carefully evaluate our work. We appreciate your insightful comments, which help us improve the clarity and strengthen the paper. We will incorporate these clarifications and additional results into the final version of the paper.
> > >
> > > Sincerely,
> > > The Authors

---

### Official Review · Reviewer_qAAr · 2026-03-12

**Soundness:** 3
**Presentation:** 3
**Significance:** 2
**Originality:** 2
**Overall Recommendation:** 4
**Confidence:** 3

**Summary:**

This paper argues that LiDAR anomaly detection has a core blind spot: most methods learn only from visible points, even though occlusion itself is a structured consequence of sensing and can carry anomaly-relevant information. The paper frames this as an identifiability issue: under a single view, missing regions caused by normal occlusion and abnormal structure are entangled, so the model cannot tell whether “nothing was seen” because of ordinary scene geometry or because an anomaly altered visibility. To address this, the paper proposes COVAL, which injects physics-conformed synthetic anomalies, constructs a factual scan and a counterfactual scan that differ only in anomaly-induced occlusion, and trains with two auxiliary objectives: VVCR, which reconstructs counterfactual features in occluded regions, and VICC, which enforces consistency on points visible in both views. Inference still uses a single factual scan.

**Compliance With Llm Reviewing Policy:**

Affirmed.

**Key Questions For Authors:**

What exactly is the auxiliary object set used in PCAC? Is it the same across all experiments, and is it comparable in external data budget to prior synthetic-anomaly baselines?

Can the authors provide a control ablation where the paired supervision is corrupted, randomized, or made non-physics-faithful, to isolate whether the gain really comes from visibility intervention rather than just extra regularization?

Can the authors report variance across random seeds? The improvements from VVCR and VICC are meaningful but not huge compared with the earlier jumps. Seed robustness would help interpret them

Missing reference: {Towards robust lidar-based perception in autonomous driving: General black-box adversarial sensor attack and countermeasures}

**Strengths And Weaknesses:**

The problem formulation is clear. The paper is not merely proposing another synthetic-anomaly recipe; it identifies a real modeling gap in LiDAR anomaly detection: partial observability is not just nuisance noise, it is part of the signal. That is a meaningful insight, and the paired “what is missing” versus “what stays the same” decomposition is conceptually clean.

The empirical section is convincing at first glance. Results are shown on two architectures, across both point-level and object-level metrics, and the paper includes ablations on the main components, hyperparameter sensitivity, and feature-scale sensitivity.

Weakness:

My main concern is that the paper’s causal and counterfactual language is stronger than the evidence currently supports. The paper writes the method as a do(visibility = ν) intervention and defines the counterfactual observation by retaining points occluded by the injected abnormal object. That is a useful privileged training construction, but it is not obviously a physically realizable LiDAR observation from the same sensor. In other words, the method may be excellent engineered supervision, but the paper has not fully justified the stronger claim that it resolves an identifiability problem in a principled causal sense.

The evaluation scope is narrower than the conceptual claims. Most of the headline evidence is on STU validation. The paper does pretrain on SemanticKITTI and Panoptic-CUDAL, but it does not really demonstrate cross-dataset generalization of the occlusion-aware idea itself. For a paper making a sensing-level argument, broader evaluation would help a lot.

There is also no real accounting of training overhead. Since the method adds paired views, reconstruction heads, alignment, and auxiliary losses, it would be useful to know the extra wall-clock, memory, and implementation burden compared with PCAC+AL alone. The paper emphasizes that inference is unchanged, which is good, but training cost is left vague.

---

> ### Author Rebuttal · Authors · 2026-03-31
>
> We thank the reviewer for the insightful comments. We clarify these points below and will update our paper accodingly.
>
> **1. On the meaning of the “do(visibility=$\nu$ )” intervention.**
>
> We would like to clarify that our use of causal language is intended as a conceptual tool rather than claiming a physically realizable LiDAR observation. Specifically, the counterfactual view is not meant to simulate an actual sensor measurement, but to introduce a controlled intervention on the visibility mechanism while keeping the underlying scene and abnormal cause fixed. This provides supervision that is fundamentally unavailable from passive single-view observations. Under this construction, the model is exposed to multiple visibility realizations of the same scene, enabling it to distinguish between (i) visibility-dependent effects (occlusion-induced missingness) and (ii) visibility-invariant structural factors. The benefit of our method thus comes from this additional supervision structure, which helps disentangle anomaly-induced structural deviations from normal occlusion patterns and leads to more precise representations of normal scenes. We will revise the wording to make this interpretation more explicit.
>
> **2. Auxiliary object set used in PCAC.**
>
> The auxiliary object set in PCAC is constructed from random object instances sampled from the “thing” classes in the training dataset. To obtain diverse and structured abnormal variants, these instances are further transformed via controlled perturbations such as random scaling, rotation, and geometric deformation. Therefore, our method is comparable to prior synthetic-anomaly approaches in terms of data budget, as we do not rely on external datasets or extra annotations beyond standard training data.
>
> **3. Control ablation for generic regularization.**
>
> To test whether the gain comes from visibility intervention rather than extra regularization, we replaced the physics-faithful visibility intervention with random masking while keeping the same reconstruction/consistency losses. This performs worse than our full method (random-mask recon: 99.85 / 0.30 / 86.72; random-mask recon+consis: 99.76 / 0.30 / 87.64 vs. full model: 99.92 / 0.13 / 89.00), indicating that the improvement is not explained by generic auxiliary losses alone.
>
> **4. Seed variance.**
>
> We thank the reviewer for this suggestion. We conduct additional experiments with multiple random seeds and report the mean ± std for three settings as follows:
>
> | Setting  | AUROC ↑   | FPR@95 ↓   | AP ↑    |
> |---------|--------|---------|-------|
> | PCAC + AL | 99.74 ± 0.09   | 0.41 ± 0.07   | 86.99 ± 0.32   |
> | + VVCR  | 99.84 ± 0.07  | 0.21 ± 0.03   | 88.54 ± 0.24   |
> | + VVCR + VICC  | 99.90 ± 0.02   | 0.14 ± 0.02   | 89.31 ± 0.34   |
>
> The variance is consistently small across all metrics, indicating that the improvements are stable and not due to randomness. We note that performance is already close to saturation in AUROC and FPR@95, making further improvements inherently small in absolute value. In this regime, gains in AP are more indicative. In particular, AP improves consistently from 86.99 → 88.54 → 89.31, which reflects better localization and separation of anomalies.
>
> Overall, these results suggest that the improvements introduced by VVCR and VICC are consistent and meaningful despite the saturated performance regime. We will include these results in the revision.
>
> **5. training overhead.**
>
> Baseline training requires approximately 30 hours and 20 GB of memory, while COVAL introduces an additional overhead of about +4 hours and +15 GB. The extra memory mainly comes from the multi-scale VVCR reconstruction heads. Importantly, these components are only used during training and are removed at inference, resulting in no additional test-time cost. We will include these details in the revision.
>
> **6. Missing reference.**
>
> Thank you for the suggestion. We will include the discussion on adversarial LiDAR perception, which is indeed relevant to robustness under sensing perturbations.

---

> > ### Author Rebuttal · Reviewer_qAAr · 2026-04-05
> >
> > I believe the authors resolved most of the concerns, given the overall usecases and impact, I will remain my weak accept rating.

---

> > > ### Author Response · Authors · 2026-04-06
> > >
> > > Dear Reviewer qAAr,
> > >
> > > Thank you for your thoughtful comments and helpful suggestions. We appreciate your time and insights!
> > >
> > > Sincerely,
> > > The Authors

---

### Official Review · Reviewer_sfyJ · 2026-03-13

**Soundness:** 2
**Presentation:** 3
**Significance:** 3
**Originality:** 2
**Overall Recommendation:** 4
**Confidence:** 4

**Summary:**

The authors strive to focus on a core issue in LiDAR anomaly detection: the identifiability problem introduced by occlusion under single-view observations. The paper argues that existing methods learn exclusively from visible measurements, while occlusion—an inherent consequence of line-of-sight LiDAR sensing—induces structured missingness that is entangled with scene geometry. As a result, anomaly-induced structural effects may be absorbed into normal variability during training. To address this, the paper proposes Counterfactual Occlusion-Visibility Anomaly Learning (COVAL). The method introduces physics-conformed synthetic abnormal perturbations and constructs paired factual and counterfactual observations that differ only in visibility realization. Extensive experiments on the STU benchmark using MinkNet and Mask4Former3D backbones show substantial improvements in both point-level and object-level anomaly segmentation metrics, achieving state-of-the-art performance.

Overall, the study's major concept concerns explicit visibility intervention as a means to disentangle anomaly-induced structural deviations from occlusion-induced partial observability in LiDAR sensing.

**Compliance With Llm Reviewing Policy:**

Affirmed.

**Key Questions For Authors:**

•	How sensitive is performance to the diversity and realism of the synthetic anomaly construction

•	What happens if visibility intervention is applied without synthetic anomaly perturbation?

•	Can the identifiability claim be formalized in a causal graph framework?

•	How much additional training time and memory does COVAL require compared to baseline fine-tuning?

•	Have you tested cross-dataset anomaly generalization (e.g., train on STU, evaluate on a different anomaly benchmark)?

•	What are the primary failure cases observed in practice?

**Limitations:**

•	The method assumes accurate modeling of occlusion under line-of-sight constraints.

•	It relies on controlled synthetic abnormal causes that may not span real-world anomaly diversity.

•	The intervention is simulated rather than derived from real multi-view LiDAR observations.

•	Broader generalization beyond STU remains unverified.

•	The causal interpretation is conceptual rather than formally validated.

**Strengths And Weaknesses:**

Strengths
1. Clear Problem Identification
The paper identifies a meaningful and underexplored issue: structured occlusion effects in LiDAR anomaly detection. The identifiability argument is intuitively compelling and well-motivated.
2. Conceptual Novelty
The use of counterfactual visibility intervention is novel in the context of 3D anomaly detection. The decomposition into:
visibility-variant modeling (reconstruction of missingness), and visibility-invariant consistency (feature alignment),is conceptually clean and complementary
3. Strong Empirical Results
Significant improvements in AUROC, FPR@95, and AP.
Large reduction in false positives compared to prior methods.

Weaknesses
1. Dependence on Synthetic Anomaly Construction

•	The method relies heavily on physics-conformed synthetic anomaly injection. While carefully designed, it remains unclear.

•	Whether improvements stem primarily from exposure to synthetic anomalies rather than visibility intervention itself.

•	How well synthetic abnormal causes approximate real-world unknown anomalies.

•	A more detailed analysis of the realism and diversity of synthetic perturbations would strengthen the paper.

2. Limited Theoretical Formalization

Although the identifiability argument is persuasive, it is largely conceptual. The paper frames the approach in a causal perspective but does not provide:

•	A formal causal graph, theoretical guarantees of disentanglement, or measurable criteria showing reduced entanglement.

•	The causal framing could be strengthened with more formal grounding.

3. Training–Inference Mismatch

During training, the model benefits from paired factual–counterfactual supervision. At inference time, only a single factual observation is available. While empirically effective, the paper does not deeply analyze:

•	Why invariance learned under intervention generalizes to real test-time anomalies.

•	Whether the model overfits to the specific intervention design.

4. Limited Cross-Dataset Generalization

Experiments are primarily conducted on STU. Although additional datasets are used for in-distribution pretraining, the anomaly evaluation remains concentrated on a single benchmark. Broader cross-dataset evaluation would strengthen the generality claim.

5. Computational Overhead Not Quantified

The method introduces:
•	synthetic scene augmentation, counterfactual visibility construction, multi-scale projection and reconstruction heads.

The paper states that reconstruction heads are removed at inference, but training cost, memory overhead, and preprocessing complexity are not reported.

---

> ### Author Rebuttal · Authors · 2026-03-31
>
> We thank the reviewer for the thoughtful comments.  We will update accordingly in the paper.
>
> **1. Dependence on Synthetic Anomaly Construction.**
>
> We thank the reviewer for this important concern.
>
> (1) Improvement from synthetic anomalies and visibility intervention.
>
> Introducing PCAC improves AUROC from 92.43 → 98.20, showing that exposure to synthetic anomalies is beneficial. Adding visibility intervention further improves performance to 99.92, indicating that intervention provides additional gains beyond synthetic supervision by explicitly disentangling anomaly-induced structural effects from normal occlusion patterns.
>
> (2) Approximation to real-world anomalies.
>
> As shown in Fig. 5, synthetic anomalies already exhibit plausible LiDAR characteristics and capture structural and visibility patterns similar to those observed in real-world anomalies (Fig. 8–10).
>
> (3) Sensitivity to realism and diversity of synthetic anomalies.
>
> We further vary the source objects from realistic categories to simple synthetic shapes. Even a simple geometric shape (Cube)  already outperforms REL, indicating that the method does not rely on semantic realism. More realistic shapes further improve performance, while using diverse classes gives the best overall balance, indicating that structural diversity is more important.
>
> | Source category| AUROC↑ | FPR@95↓ | AP↑   |
> |-|-|-|-|
> | REL | 87.82 | 40.96 | 7.26  |
> | Cube | 98.90 | 7.02 | 71.13 |
> | Single class | 99.73 | 0.87 | 85.20 |
> | All classes (paper) | 99.64 | 0.47 | 87.09 |
>
> **2. What happens if visibility intervention is applied without synthetic anomaly perturbation?**
>
> We conduct an additional ablation where visibility intervention is applied without synthetic anomaly perturbation. Compared to the full model, performance decreases when synthetic perturbations are removed. Nevertheless, it still improves over the baseline, suggesting that modeling visibility variation provides useful supervision. When combined with PCAC, it further improves performance, indicating complementary benefits, indicating that it provides complementary benefits..
> |  | AUROC ↑ | FPR@95 ↓ | AP ↑  |
> |-|-|-|-|
> | Baseline| 92.43   | 22.70    | 0.91  |
> | PCAC   | 98.20   | 12.93    | 78.49 |
> | Interv  | 96.49   | 16.79    | 12.93 |
> | Our(ML)| 98.47   | 10.90    | 82.10 |
>
> **3. Can the identifiability claim be formalized more explicitly?**
>
> Thank you for this suggestion. We agree that the causal framing can be strengthened with more formal grounding. We can make the causal structure explicit with the SCM
>
> $A \rightarrow S \rightarrow X\; A \rightarrow \nu \rightarrow X$
>
> where A is the abnormal cause, S is the normal scene geometry, $\nu$ is the visibility mechanism, and X is the observed point cloud. Under passive observation, visibility and structure are coupled in a single realization, so their effects are entangled. Our paired construction introduces controlled variation in visibility while keeping the scene and abnormal perturbation fixed, providing supervision to separate visibility-invariant structural cues from visibility-dependent occlusion effects.
>
> **4. Training overhead.**
>
> Baseline training requires about 30h and 20 GB memory, while our method introduces an overhead of approximately +4h and +15 GB. Importantly, these components are used only during training and removed at inference.
>
> **5. Cross-dataset generalization.**
>
> Indeed that broader evaluation would be more beneficial. Yet currently, STU is the only public real-world LiDAR anomaly segmentation benchmark, which limits standardized cross-dataset evaluation. Importantly, our method does not rely on dataset-specific anomaly categories, but instead learns structural deviations via synthetic perturbations and visibility intervention, which are dataset-agnostic. This suggests that the learned representation is expected to generalize beyond STU, and we will further explore this in future work.
>
> **6. Failure cases.**
>
> As shown in Tab. 7, Seq. 142 has a much lower AP compared to other sequences. This failure case is mainly associated with very small or low-height anomalies, which produce few LiDAR returns and weak occlusion cues. In such scenarios, both structural evidence and visibility contrast are limited, making these anomalies harder to distinguish from background clutter.
>
> **7. Training–inference mismatch.**
>
> We thank the reviewer for this important point. The goal of intervention is to enhance more discriminative representation learning. By forcing the model to explain "what is missing" and "what stays the same" during training for abnormal objects, the model learns a tighter boundary for what constitutes  "normal object" and "normal occlusion". This refined understanding of normal scene structure enables the model to identify deviations at test time. We will update the paper to further clarify this.

---

> > ### Author Rebuttal · Reviewer_sfyJ · 2026-04-04
> >
> > Thank you for the detailed response and additional experiments. I have no other comments.

---

> > > ### Author Response · Authors · 2026-04-04
> > >
> > > Dear Reviewer sfyJ,
> > >
> > > Thank you for your valuable feedback and careful review. We appreciate your insights and will incorporate the suggested clarifications into the final version.
> > >
> > > Sincerely,
> > >
> > > The Authors

---

### Decision · Program_Chairs · 2026-04-30

**Decision:**

Accept (regular)

**Comment:**

This paper studies an important and underexplored issue in LiDAR anomaly detection, namely the entanglement between anomaly-induced structural changes and occlusion-induced missingness under single-view sensing. The initial concerns mainly centered on the strength of the causal and identifiability claims, the reliance on synthetic anomaly construction, and the limited breadth of evaluation. After the rebuttal, all reviewers indicated that their concerns were fully resolved, and all four reviewers consistently maintained a Weak Accept recommendation. Given this overall assessment, I support Weak Accept. The authors should incorporate the promised clarifications, additional analyses, and corresponding revisions in the final version.